# Theabrownin Isolated from Pu-Erh Tea Enhances the Innate Immune and Anti-Inflammatory Effects of RAW264.7 Macrophages via the TLR2/4-Mediated Signaling Pathway

**DOI:** 10.3390/foods12071468

**Published:** 2023-03-30

**Authors:** Lei Zhao, Yue Miao, Bo Shan, Chunyan Zhao, Chunxiu Peng, Jiashun Gong

**Affiliations:** 1College of Food Science and Technology, Yunnan Agricultural University, Kunming 650201, China; 2College of Science, Yunnan Agricultural University, Kunming 650201, China; 3College of Horticulture and Landscape, Yunnan Agricultural University, Kunming 650201, China; 4Agro-products Processing Research Institute, Yunnan Academy of Agricultural Sciences, Kunming 650223, China

**Keywords:** theabrownin, RAW264.7 macrophages, innate immune enhancement, anti-inflammatory effect, TLR2/4, NF-κB/MAPK/PI3K–AKT signaling pathways

## Abstract

Theabrownin (TB) is a tea pigment extracted from Pu-erh Tea. The effects of TB on innate immunity and inflammation are not well understood. Herein, the effects of TB on innate immunity are investigated using RAW264.7 macrophages. We found that TB promoted the proliferation of RAW264.7 macrophages, altered their morphology, enhanced their pinocytic and phagocytic ability, and significantly increased their secretion of nitric oxide (NO) and cytokines, all of which enhanced the immune response. Additionally, TB inhibited the release of inflammatory signals in RAW264.7 macrophages primed with lipopolysaccharide (LPS), implying that TB modulates the excessive inflammation induced by bacterial infection. A Western blot showed that TB could activate the toll-like receptor (TLR)2/4-mediated myeloid differentiation factor 88 (MyD88)-dependent mitogen activated protein kinase (MAPK) and nuclear factor-κB (NF-κB) signaling pathway and the TLR2-mediated phosphoinositide 3-kinase (PI3K)–AKT signaling pathway, enhancing the immune functions of RAW264.7 macrophages. TB also inhibited the phosphorylation of core proteins in the MAPK/NF-κB/PI3K–AKT signaling pathway induced by LPS. In addition, we analyzed the transcriptomes of RAW264.7 macrophages, and a Kyoto encyclopedia of genes and genomes (KEGG) enrichment analysis revealed that TB modulated thetoll-like receptor signal pathway. A gene ontology (GO) enrichment analysis indicated that TB treatment strongly modulated the immune response and inflammation. As a result, TB-enhanced innate immunity and modulated inflammation via the TLR2/4 signaling pathway.

## 1. Introduction

Toll-like receptors (TLRs) were first identified in 1989 by Janeway as pattern recognition receptors (PRRs) [1]. TLRs recognize microorganisms via pathogen-associated molecular patterns (PAMP) [2] and regulate innate immunity and inflammation [3]. TLRs further activate signal transduction in immune cells, inducing the transcription and expression of cytokines that enhance cellular immune function [4]. TLR4 and TLR2 are the most important membrane receptors expressed in macrophages. The binding of corresponding ligands to TLR4/2 activates MAPK [5], c-Jun N-terminal kinase (JNK), and NF-κB signaling pathways via the MyD88-dependent pathway, which induces the secretion of cytokines and NO [6]. In addition, TLR2 enhances immunity by activating the PI3K–AKT signaling pathway [7].

Macrophages perform their functions through phagocytosis, intracellular killing activity, and cytotoxicity against invading organisms. At present, RAW264.7 macrophages are used to study natural compounds that could enhance the immune system [8,9]. These natural active products potentially have strong immunomodulatory properties and could modulate inflammation and the adverse effects of an excessive immune response [10,11]. Priming RAW264.7 macrophage with LPS is a commonly used method of inducing inflammation. LPS combines with CD14 on the macrophage membrane, activating TLR4 and stimulating the expression of numerous proinflammatory cytokines that induce inflammation [12].

Pu-erh tea is exclusively produced from the fresh leaves and buds of the large-leaf tea plant species *Camellia sinensis* var. *Assamica* [13]. Pu-erh tea has been consumed for more than 1800 years and is divided into raw and ripened tea. Raw Pu-erh tea is not fermented. However, ripened Pu-erh tea is made by carrying out the solid-state fermentation of sun-dried green tea. The numerous polyphenols in the tea undergo microbial oxidation and polymerization and are complexed with polysaccharides, theanine, theophylline, and other compounds in the tea to form theabrownin (TB) (10–12%), a dark-brown complicated product [14,15]. TB not only determines the flavor of Pu-erh tea, but also its physiological activity. Studies have shown that fermented Pu-erh tea has a better blood lipid lowering and anti-atherosclerotic effect than raw tea [16]. Several recent studies have found that TB possesses hypolipidemia [17], hypoglycemia [18], anticancer [19], and antioxidation properties [20], and that it regulates the abundance of intestinal microorganisms [21].

Pu-erh tea also lowers the LPS-induced high NO production in the livers of rats by modulating the signal transduction of TLR4 [22]. In aging mice, TB improves oxidative stress, eliminates excess free radicals in the liver, and modulates inflammation [23]. However, the ways in which TB affects the immune function and alleviates inflammation in vitro are still unclear. Understanding the functional mechanisms of Pu-erh tea could enrich our knowledge of the health value of this tea. Herein, RAW264.7 macrophages were used to study the immune enhancement and anti-inflammatory properties of TB, the underlying mechanisms, and the related signaling pathways.

## 2. Materials and Methods

### 2.1. Materials and Reagents

The ripened Pu-erh tea we used was Dayi brand, label no. 7262, produced on 13 May 2018. RAW264.7 cells were obtained from the Kunming Institute of Zoology. CAS (Kunming, China). Fetal bovine serum (FBS) (04-001-1A), Dulbecco’s modified eagle’s medium (DMEM) (01-052-1A), penicillin, and streptomycin (03-033-1B) were purchased from Biological Industries (Tel Aviv, Israel). A Vybrant Phagocytosis Assay Kit (V6694) was purchased from Thermo Fisher Scientific (Waltham, MA, USA). Tumor necrosis factor (TNF)-α (E-EL-M3063), interleukin (IL)-1β (E-EL-M0037c), and IL-6 (E-EL-M0044c) detection enzyme-linked immunosorbent assay (ELISA) kits were purchased from Elabscience Biotechnology (Wuhan, China). A Griess reagent kit (A013-2-1) was purchased from Nanjing Jiancheng Bioengineering Institute (Nanjing, China). TRNzol Universal (DP424), a FastKing RT Kit (With gDNase) (KR116), and a FastKing One Step RT-qPCR SYBR Green Kit (FP313) were purchased from TianGen (Beijing, China). A polyvinylidene fluoride (PVDF) membrane (0.45 μm) was purchased from Immobilon-P (Merck KGaA, Darmstadt, Germany). Skimmed milk powder was purchased from BD Biosciences (San Diego, CA, USA). An efficient chemiluminescence (ECL) kit (BL520A) and bicinchoninic acid (BCA) protein assay kits (BL521A) were purchased from Biosharp (Hefei, China). LPS (L8880), neutral red (N8160), RIPA buffe (R0010), bovine serum albumin (BSA) (A8020), and TBST (T1082) were purchased from Solarbio (Beijing, China). Primary antibodies TLR2 (2229S), TLR4 (14358S), Phospho-p38 (p-p38) (4511T), phospho-JNK (p-JNK) (4668T), and phospho-ERK (p-ERK) (4370T) were purchased from Cell Signaling Technology (Beverly, MA, USA). TLR4 (66350-1-Ig), MyD88 (67969-1-Ig), p38 (66234-1-Ig), JNK (66210-1-Ig), ERK (11257-1-AP), p65 (10745-1-AP), IKBα (10268-1-AP), and β-actin (66009-1-Ig) were purchased from Proteintech Group, Inc. (Rosemont, IL, USA). AKT (A11016), phospho-p65 (p-p65) (AP0123), and phospho-AKT (p-AKT) (AP0140) were purchased from ABclonal (Wuhan, China). Phospho-IKBα (p-IKBα) (AF2002) was purchased from Affinity Biosciences (Beijing, Cnina). A cell counting kit-8 (CCK-8) (PF00004), prestained protein marker (PL00001), and secondary antibodies were purchased from Proteintech Group, Inc. (Rosemont, IL, USA). The other chemicals and reagents were of analytical grade.

### 2.2. Extraction of Theabrownin

TB was extracted according to a previously described method [23], and the yield was 10.56%.

### 2.3. Cell Culture

The RAW264.7 cells were cultured in DMEM supplemented with 10% (*v*/*v*) FBS, 100 µg/mL streptomycin, and 100 units/mL penicillin in a 5% CO_2_/air environment at 37 °C. The cells were sub-cultured every 1–2 days at a density of 1 × 10^6^ cells/mL and used in all experiments between passage 8 and 15.

### 2.4. Cell Viability Assay

First, 100 μL of cells were cultured in a 96-well plate overnight. The supernatant was removed and various concentrations of TB (62.5 μg/mL, 125 μg/mL, 250 μg/mL, 500 μg/mL, 1000 μg/mL, 1500 μg/mL) were added for 24 h or 48 h. Next, 100 μL of 10% CCK8 reagent was added to every well and incubated for 1 h at 37 °C. Finally, absorbance was detected using a microplate reader (Thermo Fisher, Waltham, MA, USA) at 450 nm. The cell viability was calculated using the following equation:$$\mathrm{Cell}\;\mathrm{viability}\;\%\;=\frac{Ab_{TB}-Ab_{blank}}{Ab_{con}-Ab_{blank}}\times 100\%$$

### 2.5. Morphologic Observation

#### 2.5.1. Immune Enhancement Groups

First, 3 mL of cells were cultured in 60 mm × 60 mm culture plates for 12 h and then incubated with various concentrations of TB (0 μg/mL, 62.5 μg/mL, 125 μg/mL, 250 μg/mL, 500 μg/mL) and 1 μg/mL LPS for 8 h. Normal RAW264.7 cells were used as a control group (Con-group). The morphologies of the RAW264.7 cells were observed using an inverted microscope (Leica, Japan). The culture supernatant or cells were collected for the following experiments.

#### 2.5.2. Anti-Inflammatory Groups

First, 3 mL of cells were cultured in 60 mm × 60 mm culture plates for 12 h and then incubated with various concentrations of TB (0 μg/mL, 62.5 μg/mL, 125 μg/mL, 250 μg/mL, 500 μg/mL) for 1 h. Next, 1 μg/mL LPS was added for 8 h. Normal RAW264.7 cells were used as a control group (Con-group). The morphologies of the RAW264.7 were observed using an inverted microscope. The culture supernatant or cells were collected for the following experiments.

### 2.6. Measurement of Pinocytic and Phagocytic Capacity

#### 2.6.1. Pinocytic Capacity

First, 100 μL of cells were cultured in 96-well plates for 12 h. The cells were then incubated with various concentrations of TB or LPS (1 μg/mL) for 8 h. Next, 100 μL of 0.1% neutral red was incubated for 1 h. After washing the cells with PBS, 100 μL of 1% acetic acid solution (*v*/*v*) in 50% ethanol (*v*/*v*) was added to each well. The absorbance was recorded the next day using a microplate reader at 540 nm.

#### 2.6.2. Phagocytic Capacity

First, 100 μL of cells were cultured in 96-well plates for 12 h. The cells were then incubated with various concentrations of TB or LPS (1 μg/mL) for 8 h. Next, 100 μL of FITC-labeled *E. coli* was added to each well and incubated at 37 °C for 2 h. The bioparticle loading suspension was removed from each well and trypan blue was added at room temperature. The trypan blue was removed after 1 min. Images of RAW264.7 cells were captured using a fluorescence microscope (Leica, Japan), and the fluorescence intensity was measured using a fluorescence microplate reader (Bio-tek, Burlington, VT, USA) at 480 nm (excitation)/520 nm (emission). The efficiency of phagocytosis was calculated according to the method recommended by the manufacturer.

### 2.7. Determination of NO and Cytokines

The NO production was measured using a Griess reagent kit. The absorbance was recorded at 540 nm. A standard curve was established for the calculation of the NO.

The production of TNF-a, IL-6, and IL-1β was measured using an ELISA kit. The absorbance was recorded at 450 nm and the standard curves were established for the calculation.

### 2.8. Reverse Transcription and Real-Time Quantitative PCR

The total RNA of the cells was treated with TRNzol Universal reagent following the manufacturer’s protocol. Glyceraldehyde-3-phosphate dehydrogenase (GAPDH) was used as the internal reference. The RNA (2 μg) was reverse transcribed with a total volume of 20 μL, and the primer sequences are shown in Table 1. Gene amplification was carried out using an ABI StepOne Plus instrument (Applied Biosystems Inc., Carlsbad, CA, USA). The PCR process was as followed: 95 °C for 15 min, 40 cycles at 95 °C for 10 s, 60 °C for 30 s, and extension for 15 s at 95 °C. The 2^−ΔΔCT^ method was used to measure the relative mRNA expression levels.

### 2.9. Investigation of Membrane Receptors (TLR2/4)

First, 2 mL of cells were cultured in a 6-well culture plate for 12 h. The cells were then treated with antibodies of TLR2 and TLR4, or a mixture of these two antibodies, for 2 h. After 2 h, TB was added to the cells (125 μg/mL or 250 μg/mL) for 8 h and the cytokine secretion levels were then measured.

### 2.10. Western Blot Analysis

The proteins of all the groups were collected by the RIPA lysis buffer in an ice-bath. The protein contents were measured using BCA kits. Next, 20 μg of proteins were loaded onto 7.5% or 10% SDS–acrylamide gel by electrophoresis and semi-dry transferred to PVDF membranes. The PVDF membranes were blocked with 5% skimmed milk at room temperature for 1 h and then incubated with primary antibody overnight at 4 °C. Finally, the PVDF membranes were washed with TBST and incubated with secondary antibody for 1 h. Images of target blots were captured using ECL by ChemiDoc Touch Imaging (Bio-rad, Hercules, CA, USA) and quantified using the ImageJ software (NIH, Bethesda, MD, USA).

### 2.11. mRNA-seq and Transcriptome Analysis

The RAW264.7 cells were cultured according to the process outlined in Methods 2.5.1., and three independent samples were collected using the Trizol reagent. The transcriptome sequencing of the extracted RNA and data analysis were performed by BioTree (Shanghai, China) (http://www.biotree.com.cn/). The principal component analysis (PCA) and Venn analysis were performed according to the processes described by Lc-bio (https://www.lc-bio.cn/).

### 2.12. Statistical Analysis and Graphics Processing

All the results are expressed as mean ± SD, and an unpaired, one-tailed Student’s t-test was performed for the comparison within the groups using GraphPad Prism (version 8.0.2). Differences between the groups were considered significant (*p* < 0.05). Bar charts were drawn using GraphPad Prism (version 8.0.2). The results of Western blotting in the graphs are presented as the mean ± SD from three independent measurements using the ImageJ program.

## 3. Results

### 3.1. Effects of TB on RAW264.7 Cell Viability

RAW264.7 cells were treated with various concentrations of TB (0 μg/mL, 62.5 μg/mL, 125 μg/mL, 250 μg/mL, 500 μg/mL, 1000 μg/mL, 1500 μg/mL) for 24 h or 48 h (Figure 1A,B). For the 24 h treatment, 1000 μg/mL TB significantly inhibited cell viability (48.313%). The lower and moderate TB concentrations (62.5 μg/mL, 125 μg/mL, 250 μg/mL) still had a significant effect on the proliferation of the RAW264.7 cells (134.07%, 126.22%, 117.38%, respectively). The trend in the viability of the RAW264.7 cells under the 48 h treatment was comparable to that of the 24 h treatment. Therefore, 62.5 μg/mL, 125 μg/mL, 250 μg/mL, and 500 μg/mL concentrations of TB were used in the subsequent experiments.

### 3.2. Effect of TB on RAW264.7 Morphology

The effect of the TB on the morphology of the RAW264.7 macrophages is shown in Figure 1C–L. The morphology of the cells in the Con-group was mostly round, but a few elongated elliptical cells were observed (Figure 1C). However, culture in 1 μg/mL LPS altered the morphology of the cells, resulting in obvious tentacles or branches around the cells, some of which were radial (Figure 1H). These morphological changes increased the surface area of the RAW264.7 cells and enhanced their phagocytosis ability (a normal inflammatory reaction of macrophages) [24]. Compared with the cells in the Con-group, the morphologies of the RAW264.7 cells in each TB group were slightly altered, with less branching and deformation (Figure 1D–G). This indicated that TB could stimulate a stress response in RAW264.7 cells, but that the response was completely different from that induced by LPS. However, culturing in TB restored the morphology of the RAW264.7 cells altered by LPS. Based on the morphologies we observed, TB stimulates RAW264.7 cells and enhances their immune response ability. Notably, the stimulation of TB is sufficient but not excessive, and TB further modulates the over-stimulation and inflammatory reaction of RAW264.7 cells.

### 3.3. TB Enhances the Phagocytic Capacity of Macrophages

In the host defense system, phagocytes are the first cells to respond to pathogenic microorganisms, underscoring their importance in the innate immune response. Neutral red is used to study pinocytic capacity, i.e., the ability of macrophages to ingest small molecules. The effect of TB on pinocytosis is shown in Figure 1M. Compared with the cells in the Con-group, 62.5 μg/mL TB significantly increased the pinocytosis ability of the macrophages. In particular, 250 μg/mL TB was the optimal dose for the highest pinocytosis. When the TB concentration was 500 μg/mL, the pinocytosis decreased. The ability of RAW264.7 macrophages to phagocytize macromolecules (fluorescent labeled *E. coli*) under TB intervention (Figure 1O–T) was also investigated. Combining these two properties is one of the most effective ways of evaluating the phagocytic ability of RAW264.7 macrophages. The uptake of *E. coli* by cells in the control group was low. The lower to medium concentrations of TB significantly increased the phagocytosis by the macrophages. The phagocytosis effect was highest at 250 μg/mL TB, but was low at 500 μg/mL. The fluorescence intensity of the cells was quantitatively analyzed using a microplate reader (Figure 1N). The fluorescence intensity was 3.44 times as high in the LPS treatment group as in the control group, and in the 62.5 μg/mL, 125 μg/mL, and 250 μg/mL TB treatment groups it was 1.10, 1.26 and 1.51 times higher, respectively, than in the control group.

TB effectively enhances the phagocytic capacity of RAW264.7 macrophages, but only within certain limits. Over-activation of macrophages leads to excessive inflammation. LPS is one of the compounds that can over-activate macrophages and trigger inflammation [25]. Based on our findings, low and medium concentrations of TB activate RAW264.7 macrophages and enhance innate response, but do not cause an excessive immune response.

### 3.4. Immune-Enhancing Effects of TB on NO and Cytokine Production

Activated RAW264.7 macrophages secrete NO and cytokines that could enhance innate immunity. However, excessive secretion of the NO and cytokines causes excessive inflammation [26]. The secretion of NO and cytokines was analyzed using the Griess method and ELISA kits (Figure 2A–D). The results showed that various concentrations of TB obviously increased the levels of NO, IL-6, IL-1β, and TNF- α. For instance, compared with the Con-group, 250 μg/mL TB increased the secretion of NO, IL-6, IL-1β, and TNF-α by 2.41, 2.79, 1.68, and 1.86 times, respectively. The low to medium TB concentrations significantly increased the expression of NO and the cytokines mentioned above. However, the high TB concentration (500 μg/mL) produced the opposite effect. Compared with the Con-group, the LPS-stimulated RAW264.7 cells over-secreted NO, IL-6, IL-1β, and TNF-α by 10.69, 9.75, 2.10, and 4.46 times, results significantly higher than those obtained from the TB groups.

QRT-PCR was used to assess the effect of TB on the mRNA expression of iNOs and cytokines in order to further validate the effect of TB on inflammation (Figure 2A–D, right Y axis). INOs are key enzymes that regulate NO production. The expression of iNOs, IL-6, IL-1β, and TNF-α mRNA were notably higher in the TB treatment groups, consistent with the ELISA results. For instance, compared with the Con-group, 250 μg/mL TB increased the expression of iNOs, IL-6, IL-1β, and TNF-α mRNA by 1.16, 1.16, 1.28, and 2.95 times, respectively.

### 3.5. Anti-Inflammatory Effects of TB on NO and Cytokine Production

Inflammation is a basic response of the immune system, and it helps protect the body from infection and tissue damage [27]. However, over-activation of macrophages induces the excessive secretion of pro-inflammatory molecules that can damage host cells and tissues [28]. Thus, the anti-inflammatory effect of TB was explored. The results indicated that TB had a dual effect on RAW264.7 macrophages.

TB significantly inhibited the abnormal production of NO and cytokines in RAW264.7 macrophages treated with LPS (Figure 2E–H, left Y axis). The effect was greater for higher TB concentrations. Specifically, compared with the LPS group, 250 μg/mL TB reduced NO, IL-6, IL-1β, and TNF-α expression by 84.24%, 82.87%, 48.89%, and 74.26%, respectively. In addition, the QRT-PCR results showed that TB treatment decreased the transcription of iNOs, IL-6, IL-1β, and TNF-α mRNAs, a result consistent with its effect on secretion.

### 3.6. TLRs of TB on RAW264.7 Macrophages

The induced expression of TLRs is one of the molecular mechanisms of immune enhancement [29]. There is a lot of TLR2/4 on the cell membrane of RAW264.7 macrophages, which makes them a good model for studying the TLR signaling pathway [30]. TLR2 antibodies (anti-TLR2) and TLR4 antibodies (anti-TLR4) could specifically bind to TLR2/4 on the RAW264.7 cell membrane, blocking the related TLR signaling pathway [31]. Based on the previous experimental results, the immune enhancement effect of TB was optimal at low and medium concentrations. Thus, 125 μg/mL and 250 μg/mL TB were selected for the antibody blocking experiment.

The results showed that blocking TLR2 or TLR4 with corresponding antibodies significantly modulated the secretion of NO and cytokines. The effect of blocking both TLR2 and TLR4 was superior to blocking either receptor (Figure 2I–L). These results imply that TB binds TLR2 and TLR4, activating intracellular signal transduction. Furthermore, the blocking of TLR2 and TLR4 down-regulated the LPS-induced activation of RAW264.7 cells, though not entirely, suggesting that LPS activates RAW264.7 cells via other pathways. 

### 3.7. TB Enhances the Immune Function of RAW264.7 Cells via the TLR2/4-Mediated MAPK/NF-κB/PI3K–AKT Signaling Pathway

Activation of TLR2 and TLR4 induces signal transduction via the MyD88 pathway, which further activates the downstream NF-κB and MAPK pathways [32]. TLR2 directly activates the PI3K–AKT signaling pathway [33]. The NF-κB/MAPK/PI3K–AKT signaling pathways enhance the innate immunity or modulate inflammation.

A Western blot was used to further study the effects of TB on the membrane receptors TLR2/4 of RAW264.7 macrophages and the transmission of intracellular signals. Both TB and LPS increased the expression of TLR2 and TLR4 (Figure 3A). Compared with the Con-group, 62.5 μg/mL, 125 μg/mL, 250 μg/mL, and 500 μg/mL TB increased the TLR2 expression by 1.08, 1.45, 1.46, and 1.45 times and the TLR4 expression by 1.42, 1.97, 2.36, and 2.06 times, respectively, indicating that TB can activate TLR2/4 at the same time. In addition, LPS increased the expression of TLR2 by 1.67 times and the expression of TLR4 by 2.45 times compared with the Con-group. Interestingly, the MyD88 content correlated with the expression of TLR2/4. TB concentrations of 62.5 μg/mL, 125 μg/mL, 250 μg/mL, and 500 μg/mL increased the expression of MyD88 by 1.25, 1.86, 1.58, and 1.23 times compared with the Con-group (Figure 3A). Similarly, TB increased the expression levels of p-JNK, p-p38, and p-ERK in the RAW264.7 cells, but had no effect on the expression of JNK, p38, or ERK (Figure 3B). TB activated IκBα, and the level of p-IκBα was obviously high. TB significantly increased the expression of p-p65 (Figure 3C). In addition, TB activated the PI3K–AKT signaling pathway by stimulating the dimer of TLR2 and TLR1, and it also promoted the phosphorylation of AKT (Figure 3D).

The Western blot results suggest that TB treatment activates the TLR2/4-mediated MAPK/NF-κB signaling pathway through the MyD88-dependent pathway and activates the PI3K–AKT signaling pathway via TLR2, enhancing the immune response of RAW264.7 cells.

### 3.8. Anti-Inflammatory Effect of TB on the TLR2/4-Mediated MAPK/NF-κB/PI3K–AKT Signaling Pathway of RAW264.7 Cells

Enhancing the immune function could inhibit the development of inflammation. The results outlined in the previous sections indicate that TB enhances the innate immunity of RAW264.7 cells, while LPS activates an inflammatory reaction. Many bioactive substances inhibit LPS-induced inflammation via the TLR2/4 pathway [34,35]. A Western blot was used to study the anti-inflammatory mechanisms of TB. Several pathways, including the TLR2/4-mediated MyD88-dependent pathway, as well as the phosphorylation of the key proteins in the MAPK, NF-κB, and PI3K–AKT signaling pathways were investigated.

The results showed that TB inhibited the overexpression of TLR2/4 and MyD88 in LPS-induced RAW264.7 cells (Figure 4A). We speculate that TB reversed or blocked the effects of LPS on TLR2/4. The effects of moderate to high TB concentrations on MyD88 were superior to those of lower concentrations. The expression of the key proteins in the MAPK signal pathway was also quantified. Compared with the LPS treatment group, the expression of p-ERK, p-p38, and p-JNK were significantly lower in the TB groups (Figure 4B). In addition, TB significantly inhibited the phosphorylation levels of p65 and IκBα in the NF-κB signaling pathway, stimulated by LPS (Figure 4C). Moreover, TB inhibited the high phosphorylation of AKT in the PI3K–AKT signal pathway (Figure 4D).

In summary, TB had a good inhibitory effect against LPS-induced over-activation of RAW264.7 cells, and this was achieved through negative regulation of TLR2/4-mediated expression of NF-κB, MAPK, and PI3K–AKT.

### 3.9. Effect of TB on Transcriptomics of RAW264.7 Cells

A principal component analysis (PCA) of the RNA-seq data of the RAW264.7 cells cultured with various concentrations of TB and LPS revealed that the effect of TB on transcriptome was completely different from that of LPS (Figure 5A), although they could produce some similar effects, such as cytokine secretion or the promotion of phagocytosis. The difference in the number of differential genes between the TB group and the Con-group was not significant. Compared with the Con-group, there were 224 genes down-regulated and 58 genes up-regulated in the 250 μg/mL TB group (Figure 5B). The number of differently expressed genes gradually rose as the TB concentration increased, and it was highest at the 500 μg/mL concentration (Figure 5C). Further analyses of the TB groups and LPS group revealed that lager numbers of differently expressed genes increased as the TB concentration increased (Figure 5D). Compared with the LPS group, 1404 genes were down-regulated and 899 genes were up-regulated in the 250 μg/mL TB group. Figure 5E,F shows a volcano map and heat map of these differently expressed genes (250 μg/mL TB vs. LPS). A KEGG enrichment analysis showed that the toll-like receptor signal pathway was among the most affected pathways. The TNF signaling, cytokine–cytokine receptor interaction, NOD-like receptor signaling, and IL-17 signaling pathways were among other important signaling pathways regulated by the TB, and they are all related to immunity and inflammation (Figure 5G). A GO analysis indicated that TB treatment strongly affected the response to bacterium, the inflammatory response, the immune and innate immune response, the immune system process, the defense response, and the response to virus of the RAW264.7 cells (Figure 5H). Future follow-up research will delve deeper into other pathways and biological processes related to immune responses regulated by TB.

## 4. Discussion

TB is a polyhydroxylated compound rich in carboxyl, hydroxyl, benzene rings, and methyl [36]. It is a high polymer with a molecular weight range of 3.5 kDa to 100 kDa [37]. Currently, polysaccharides are the most widely studied natural active compounds that enhance innate immunity [38,39]. However, tea pigments, such as TB, have been less studied. Molecular docking has revealed that the hydroxyl groups of polysaccharides are hydrogen bonded to the amino, hydroxyl, and imidazole groups of the amino acid residues of TLR2/4. TB contains an enormous number of carboxyl hydroxyl and phenolic hydroxyl groups [38], and we speculate that TB can bind with the amino acid residues of TLR2/4 by hydrogen bonding. In addition, antibody blocking assays yielded the same results, suggesting that TB may be an effective stimulant of TLR2/4 and activate its biological activity.

Immune response is an important physiological process that includes both innate and adaptive immunity. Innate immunity is the first line of defense against invading pathogens. Macrophages participate in the recognition, phagocytosis, and degradation of pathogens, infected cells, debris, and dead cells, and are the first responders to infection. Macrophages also present antigens to T cells and induce other antigen-presenting cells to play a role, thus initiating adaptive immune responses. Furthermore, at the early stages of infection, macrophages recruit other immune cells to the site of inflammation through the release of cytokines and chemokines [40]. In this study, the addition of TB altered the morphology of RAW264.7 macrophages, enhanced phagocytosis, promoted NO and cytokine production, and enhanced innate immunity function.

The immunoregulatory role of RAW264.7 macrophages is closely related to the TLRs on their cell membranes. TLRs are defined PRRs that trigger the induction of innate immune responses upon the recognition of pathogens. Moreover, TLRs not only recognize the initial infection specifically, but also link innate immunity to adaptive immunity by regulating the activation of antigen-presenting cells and key cytokines [41]. Thus, the TLR signaling pathway has become an important focus for studying immunity and inflammation. We focus here on the MyD-88 dependent downstream signaling pathway in the TLR signaling pathways. The MAPK signaling pathway is downstream of the MyD88-dependent pathways. JNK, ERK, and p38 are the key proteins in the MAPK pathway. The phosphorylation of these proteins activates nuclear transcription factors and plays a role in regulating the proliferation of cells, the secretion of cytokines, and phagocytosis [42]. Similarly, the NF-κB pathway is one of the downstream pathways of the MyD88-dependent pathways. The activation of NF-κB (p65) requires IκBα phosphorylation. When activated p65 enters the nucleus and is phosphorylated (p-p65) by nuclear kinase, it induces the transcription of immune-related genes, promotes cytokine production, and enhances the innate and adaptive immunity [43]. PI3K is a lipid kinase that regulates key physiological and pathological cellular activities. PI3K regulates the proliferation, migration, and differentiation of cells, and it regulates programmed death by activating protein kinase B (PKB or AKT for short) [33]. AKT is the core protein in the PI3K–AKT signaling pathway, and the phosphorylation of AKT further activates downstream regulatory pathways, including the NF-κB pathway. In this study, the addition of TB was found to effectively activate the three signaling pathways mediated by TLR2/4, increasing the phosphorylation level of key proteins and stimulating macrophage-secreting cytokines and thus enhancing innate immunity.

One concern we had was whether TB may cause excessive activation and inflammation of RAW264.7 cells, as does LPS. We therefore analyzed the transcriptomes of RAW264.7 macrophages. LPS affects 1341 genes, 814 of which are up-regulated and 527 of which are down-regulated. LPS greatly affects the transcriptional population of macrophages, causing cell inflammation and overexpression, or inducing cell damage and death. The inclusion of 250 μg/mL TB change only 282 genes, 58 up-regulated and 224 down-regulated. The GO analysis was enriched in the regulation of immune and inflammatory functions. PCA may also show fundamental differences between the two groups. TB can induce cytokine expression and increased phagocytosis, but it does not cause over-activation and inflammation.

Further studies have confirmed this, and for LPS-induced macrophage inflammation, we found that TB modulates the TLR signaling pathway and suppresses excessive inflammation. Inflammation response is a basic pathological state in infected or injured tissues and is induced by a variety of pro-inflammatory factors. However, the addition of TB successfully reduced the inflammatory signals in LPS-activated RAW264.7 macrophages. Some research has suggested that the suppression of the TLRs-MyD88 signaling pathway is an effective approach to relieving inflammation [44]. The Western blot results also verified the anti-inflammatory mechanism of TB, which negatively regulates the TLR signaling pathway activated by LPS and downregulates the TLR2, TLR4, and MyD88 content as well as the phosphorylation levels of key proteins in the MAPK/NF-κB/PI3K–AKT signaling pathway.

There is a close relationship between the immunoenhancing and anti-inflammatory properties of TB. TB enhances its own immune response to increase resistance to pathogens, and this also promotes an anti-inflammatory response. In addition, strong anti-inflammatory activity increases immunity to pathogens and reduces damage to the body. Subsequent studies will continue to investigate other signaling pathways of the KEGG enrichment analysis in order to obtain more comprehensive results concerning the mechanism of action of TB.

## 5. Conclusions

In this study, it was first found that TB modulates the immunity of RAW264.7 macrophages and inhibits inflammation. TB enhanced innate immunity by stimulating the proliferation of RAW264.7 cells and up-regulating the secretion of NO, TNF-α, IL-6, and IL-1 β. Further analysis revealed that TB may activate intracellular signaling by specifically binding to TLR2 and TLR4. Western blot results showed that TB not only up-regulates the expression of TLR2 and TLR4, but also activates the NF-κB and MAPK signaling pathways through the MyD88-dependent pathway. Furthermore, TB directly activates the PI3K–AKT signaling pathway via TLR2. In addition, TB effectively inhibited LPS-induced inflammation in RAW264.7 cells through the negative regulation of downstream pathways, mediated by TLR2/4, and inhibited the NF-κB/MAPK/PI3K–AKT signaling pathways. Finally, a GO enrichment analysis of the transcriptome data from the RAW264.7 cells revealed that TB effectively modulates the immune system and excessive inflammation. To summarize, TB is a promising biomacromolecule with immunomodulatory and anti-inflammatory properties. As such, it could be used as a functional food and pharmaceutical supplement.

## Figures and Tables

**Figure 1 foods-12-01468-f001:**
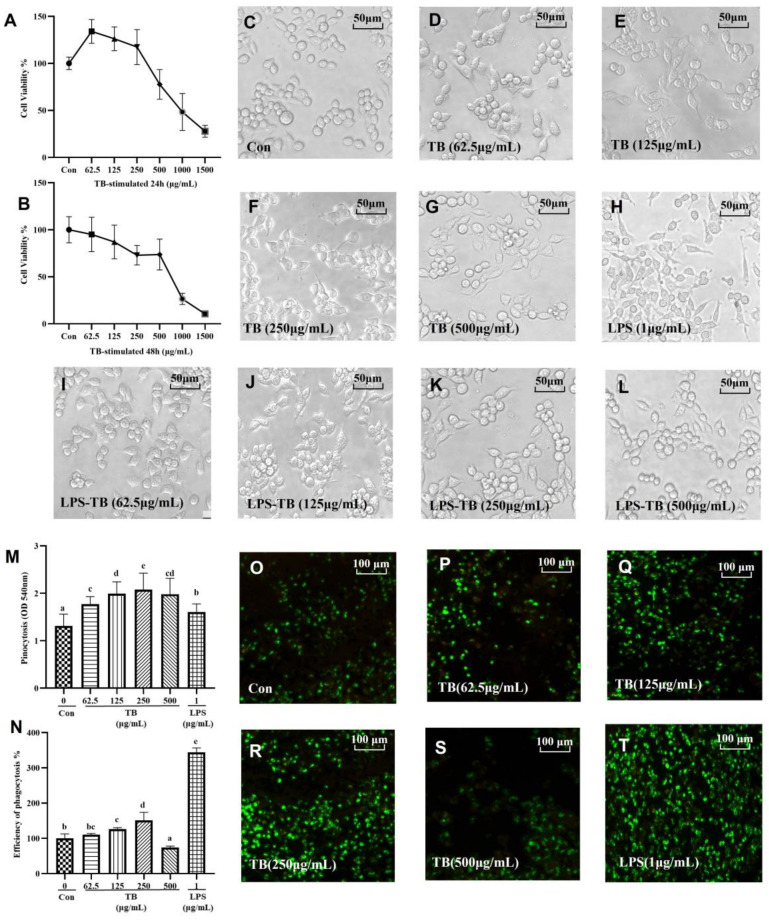
Effect of TB on RAW264.7 macrophages. (**A**,**B**) Cell viabilities (24 h and 48 h). (**C**–**H**) Effect of TB on the morphology of RAW264.7 cells (×200). (**I**–**L**) Effect of TB on the LPS-induced morphology of RAW264.7 cells (×200). (**M**) Pinocytosis effect of TB on taking neutral red. (**N**) Phagocytosis effect of TB on taking FITC-labeled *E. coli*. (**O**–**T**) Fluorescence microscopic images of phagocytosis effect of TB (×100). The statistically significant differences are marked with different letters (a, b, c, d, e).

**Figure 2 foods-12-01468-f002:**
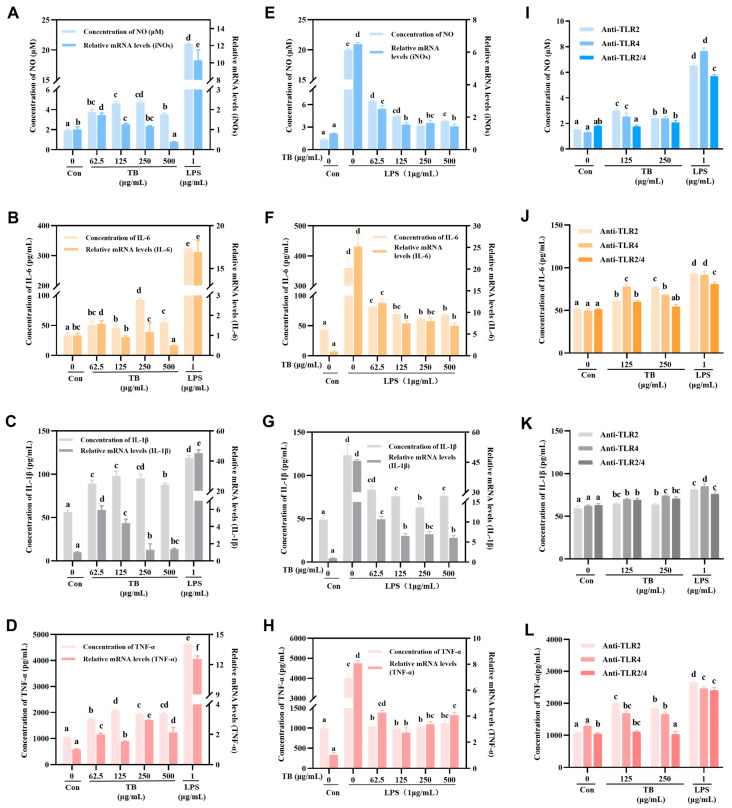
Effects of TB on production of NO and cytokines (IL-6, IL-1β, and TNF-α). (**A**–**D**) The content and mRNA expression of RAW264.7 cells from secreting NO and cytokines following TB treatment. (**E**–**H**) The content and mRNA expression of LPS-induced RAW264.7 cells from secreting NO and cytokines following TB treatment. (**I**–**L**) The content of RAW264.7 cells from secreting NO and cytokines following anti-TLR2, anti-TLR4, or anti-TLR2/4 treatment. The statistically significant differences are marked with different letters (a, b, c, d, e, f).

**Figure 3 foods-12-01468-f003:**
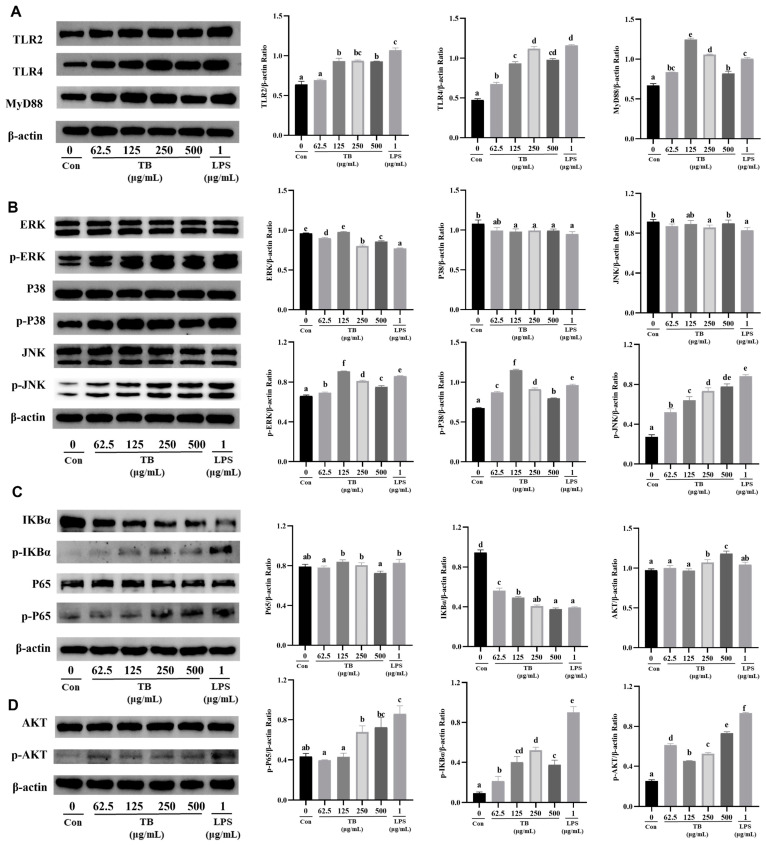
Western blot of immune enhancement effects of TB on TLR2/4-mediated pathway in RAW264.7 cells. (**A**) The expressions of TLR2/4 and MyD88. (**B**) The expressions of key proteins of the MAPK signaling pathway. (**C**) The expressions of key proteins of the NF-κB signaling pathway. (**D**) The expression of AKT and p-AKT of the PI3K–AKT signaling pathway. β-actin was used as a loading control. The statistically significant differences are marked with different letters (a, b, c, d, e, f).

**Figure 4 foods-12-01468-f004:**
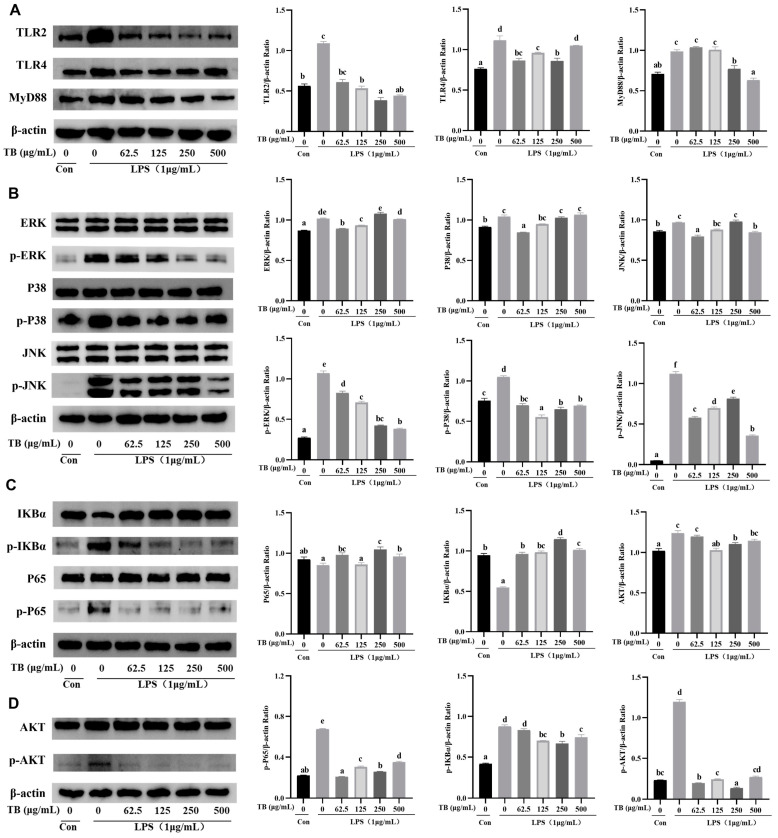
Western blot of anti-inflammatory effects of TB on TLR2/4-mediated pathway in RAW264.7 cells. (**A**) The expressions of TLR2/4 and MyD88. (**B**) The expressions of key proteins of the NF-κB signaling pathway. (**C**) The expressions of key proteins of the MAPK signaling pathway. (**D**) The expressions of key proteins of the PI3K–AKT signaling pathway. β-actin was used as a loading control. The statistically significant differences are marked with different letters (a, b, c, d, e, f).

**Figure 5 foods-12-01468-f005:**
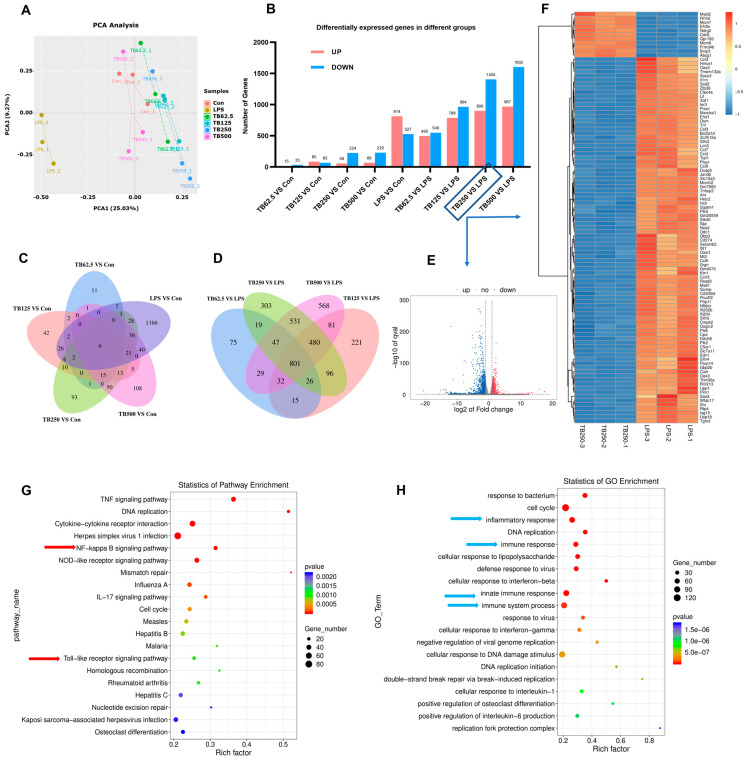
Results of cellular transcriptomics analysis. Data were extracted from RNA-seq results (three replicates per group). (**A**) PCA analysis of all groups. (**B**) Differentially expressed genes in different groups. (**C**) Venn analysis comparing genes in TB treatment groups vs. Con-group. (**D**) Venn analysis comparing genes in TB treatment groups vs. LPS group. (**E**) Volcano plots of RNA-seq data for 250 μg/mL TB group vs. LPS group. (**F**) Heat map depicting the significantly differentially expressed genes for 250 μg/mL TB group vs. LPS group. (**G**) KEGG pathway enrichment based on differential genes in 250 μg/mL TB group and LPS groups (red arrow: NF-κB and toll-like receptor signaling pathways). (**H**) GO enrichment based on differential genes in 250 μg/mL TB group and LPS groups (blue arrow: functions related to inflammation and immunity).

**Table 1 foods-12-01468-t001:** Primer sequences of targeted genes and GAPDH.

Gene	Primer Sequence	
GAPDH	Forward	5′-GCAGTGGCAAAGTGGAGATT-3′
	Reverse	5′-CGCTCCTGGAAGATGGTGAT-3′
Nitric oxide synthase (iNOs)	Forward	5′-CTTGGAGCGAGTTGTGGATTGTC-3′
	Reverse	5′-AATGTCCAGGAAGTAGGTGAGGGCT -3′
TNF-α	Forward	5′-AAAAGCAAGCAGCCAACCAG-3′
	Reverse	5′-GCCACAAGCAGGAATGAGAA-3′
IL-6	Forward	5′-CCATCTCTCCGTCTCTCACC-3′
	Reverse	5′- AGACCGCTGCCTGTCTAAAA-3′
IL-1β	Forward	5′-TGAAGGGCTGCTTCCAAACCTTTGACC-3′
	Reverse	5′-TGTCCATTGAGGTGGAGAGCTTTCAGC-3′

## Data Availability

All data are contained within the article.

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
