# Peer review of "Theabrownin Isolated from Pu-Erh Tea Enhances the Innate Immune and Anti-Inflammatory Effects of RAW264.7 Macrophages via the TLR2/4-Mediated Signaling Pathway"

_foods, 2023, doi:10.3390/foods12071468_

Round 1

Reviewer 1 Report

The authors have presented a well-crafted study that shows immense potential of Theabrownin derived from Tea in eliciting the innate immune response of macrophages through TLR2/4 mediated signaling pathway and it would certainly add interest to the readers. This manuscript has few typographical errors were observed and Catalog number of reagents must be used in the manuscript. Also, a more elaboration on method part is required especially for the "mRNA-seq and Transcriptome Analysis". Also, for western blot analysis authors have not provided information about the treatment in the method section.

Author Response

Dear Reviewer:

Thank you for your kind comments on our paper. Following your suggestions, we have supplemented several contents here and corrected several errors in our previous draft.  Based on your comments, we have also attached a letter to you. Detailed point-by-point responses are listed below:

  1. We have completed a supplementary catalog of the reagents used.
  2. ​We have explained and supplemented more in the Methods section, such as Q-PCR and WB.
  3. The experimental step of transcriptome sequencing was performed by BioTree (Shanghai, China) (http://www.biotree.com.cn/). The measurement method is: Total RNA was extracted using Trizol reagent (thermofisher, 15596018) following the manufacturer's procedure. The total RNA quantity and purity were analysis of Bioanalyzer 2100 and RNA 6000 Nano LabChip Kit (Agilent, CA, USA, 5067-1511), high-quality RNA samples with RIN number > 7.0 were used to construct sequencing library. After total RNA was extracted, mRNA was purified from total RNA (5ug) using Dynabeads Oligo (dT) (Thermo Fisher, CA, USA) with two rounds of purification. Following purification, the mRNA was fragmented into short fragments using divalent cations under elevated temperature (Magnesium RNA Fragmentation Module (NEB, cat. e6150, USA) under 94℃ 5-7min). Then the cleaved RNA fragments were reverse-transcribed to create the cDNA by SuperScript™ II Reverse Transcriptase (Invitrogen, cat.1896649, USA), which were next used to synthesise U-labeled second-stranded DNAs with E. coli DNA polymerase I (NEB, cat.m0209, USA), RNase H (NEB, cat.m0297, USA) and dUTP Solution (Thermo Fisher, cat. R0133, USA). An A-base was then added to the blunt ends of each strand, preparing them for ligation to the indexed adapters. Each adapter contained a T-base overhang for ligating the adapter to the A-tailed fragmented DNA. Dual-index adapters were ligated to the fragments, and size selection was performed with AMPureXP beads. After the heat-labile UDG enzyme (NEB, cat.m0280, USA) treatment of the U-labeled second-stranded DNAs, the ligated products were amplified with PCR by the following conditions: initial denaturation at 95℃ for 3 min; 8 cycles of denaturation at 98℃ for 15 sec, annealing at 60℃ for 15 sec, and extension at 72℃ for 30 sec; and then final extension at 72℃ for 5 min. The average insert size for the final cDNA librarys were 300±50 bp. At last, we performed the 2×150bp paired-end sequencing (PE150) on an Illumina Novaseq™ 6000 following the vendor's recommended protocol. The analysis process could be seen in the file RNA-Seq_ method. pdf.

The reason for not including this method in the paper is that it would greatly increase the repetition rate of the article. For instance, the following reference also do not directly include transcriptome sequencing methods.  

Wu J, Yeung SJ, Liu S, Qdaisat A, Jiang D, Liu W, Cheng Z, Liu W, Wang H, Li L, Zhou Z, Liu R, Yang C, Chen C, Yang R. Cyst(e)ine in nutrition formulation promotes colon cancer growth and chemoresistance by activating mTORC1 and scavenging ROS. Signal Transduct Target Ther. 2021 May 28;6(1):188. doi: 10.1038/s41392-021-00581-9. (IF=38)

  1. The analysis method of the Western Blot analysis is briefly supplemented, while the data analysis is mainly performed using ImageJ software, with the data processing and analysis being placed in section 2.12. Statistical Analysis and Graphics Processing.

The above is my response to your comments. I must again thank you for your generous help.

Best Wishes!

Dr. Lei Zhao 

Reviewer 2 Report

The authors present a work titled Theabrownin Isolated from Pu-erh Tea enhances the Innate-Immune and Anti-inflammatory Effects of RAW264.7 Macrophages via the TLR2/4-mediated Signaling Pathway" In my opinion, the work is commendable, well written and organized. The experimental section is well detailed, so that the experiments can be easily reproduced. The discussion is extensive and the cited literature is congruous with the aims of the work. In order to improve the general quality of the manuscript, I report below only some suggestions/corrections:

Please, precise the part of the plant (leaves or other) in the abstract, introduction…

Ln 50: Please, pay attention to the acronym ECG: it should be epicatechin-3-gallate

Ln 155: Please consider in the discussion that the determination of NO is of an indirect type, and comes from the direct determination of NO2 and NO3 which are also affected by the concentrations deriving from the diet.

Author Response

Dear Reviewer:

Thank you for your kind comments on our article.  Following your suggestions, we have supplemented several contents here and corrected several errors in our previous draft.  Based on your comments, we have also attached a letter to you.  Detailed point-by-point responses are listed below:

  1. We added this sentence in the introduction: The Pu-erh tea is exclusively produced from the fresh leaves with one bud and two or three leaves of large-leaf tea plant species (Camellia sinensis assamica).
  2. Thanks for the reminder. We have already noted this error. However, to reduce the repetition rate, we have removed this part of the literature and included it in the previous sentence on the immune-enhancing effects of biologically active substances.
  3. There are two pieces of evidence in the paper that can address the issue of NO detection methods. First, we performed QRT-PCR to verify that the NO production is due to the effect of TB on the activation of transcription to synthesize more nitric oxide synthase (iNOs), thus increasing the NO content. The mRNA expression of iNOs is essentially consistent with NO secretion. Of course, as an organic macromolecule, TB may be used as a food source to generate NO2 and NO3. Therefore, the second favorable evidence in this paper is that in the antibody blocking experiment, when we simultaneously block the TLR2 and TLR4 of the RAW264.7 cells, there is no significant difference of the NO secretion between the Con-group and the TB 125 μg/mL group, which indicates that the addition of TB does not cause the production of NO2 and NO3, resulting in an increase in NO during detection.

    Thank you for raising this important issue that we have overlooked. In the future study, when using the Griess method to measure NO, we will add a set of TB-solution for NO measurements (without cells), and compare them with water to ensure that our biologically active substances do not produce other NO2 and NO3 to affect the measurement of NO.

    However, from the perspective of the overall nature of this article, the Griess method for NO measurement is currently a relatively common type of method. It would be more difficult to focus on the efficacy and mechanism of TB that we are concerned with, if the principles of NO measurement and the related interpretation in this paper were discussed in a larger space. Therefore, it is hoped that you will agree that we would not include the NO measurement in the discussion of this paper. Of course, if you feel that this section must be included in the discussion, we will definitely reconsider how to revise the discussion.

The above is my response to your comments. I must again thank you for your generous help.

Best Wishes!

Dr. Lei Zhao

Reviewer 3 Report

In this study the effect of theabrownin on innate immunity was analyzed using RAW264.7 macrophages. Theabrownin is a complex product extracted from ripened Pu-erh tea which is produced from Camellia sinensis var. assamica. The results showed that theabrownin modulated the immune system by stimulating the proliferation of RAW264.7 cells and up-regulating the secretion of NO, TNF-α, IL-6, and IL-1 β, thus enhancing the innate immunity. Moreover, theabrownin inhibited LPS-induced inflammation of RAW264.7 through binding to toll like receptors 2/4 and inhibiting the phosphorylation of key proteins in NF-κB, MAPK, and PI3K-AKT signaling pathways. 

The topic is important, the manuscript is well written and provides a comprehensive analysis of the subject concluding that theabrownin could be a promising bioactive compound with immunomodulatory and anti-inflammatory effects. In summary, the title emphasized the content of the study, the abstract included sufficient information to stand alone, the introduction summarized the current state knowledge and justifies the experiment, the methods were reported with sufficient details, the results were accurately presented, with relevant data presented in figures and tables, in discussion section the findings and outcomes were discussed and correlated with previously published studies. 

I have no hesitation in recommending this manuscript after a few minor details have been attended to:

Acronyms/Abbreviations should be defined the first time they appear in the abstract, the main text, and the first figure or table 

Subsections 2.5.1. and 2.5.2. can be combined?

Line 324: “1.42, 1.97, 2.36, 2.06 folds, respectively, indicating...”

Line 327: “1.25, 1.86, 1.58, 1.23 folds, respectively (Figure.3A).”

Line 388: “lager”?

Line 461: “whether TB, like LPS, causes...”?

Author Response

Dear Reviewer:

Thank you for your kind comments on our article.  Following your suggestions, we have supplemented several contents here and corrected several errors in our previous draft.  Based on your comments, we have also attached a letter to you.  Detailed point-by-point responses are listed below:

  1. We have defined all the abbreviations that first appeared.
  2. ​Subsection 2.5.1 and 2.5.2 cannot be combined.

Subsection 2.5.1 deals with the immune activation of TB on normal RAW264.7 cells, while LPS is a positive control. In Section 2.5.2, the anti-inflammatory effect of TB on LPS-induced inflammatory cells was investigated. The purpose and steps of the two experiments are different.

  1. Line 324: “1.42, 1.97, 2.36, 2.06 folds, respectively, indicating...” I have corrected these sentences because there were some grammatical errors before. “Compared with the Con-group, LPS increased the expression of TLR2 by 1.67 folds and TLR4 by 2.45 folds. Meanwhile, 62.5, 125, 250, 500 μg/mL TB increased the expression of TLR2 by 1.08, 1.45, 1.46, 1.45 folds and TLR4 by 42, 1.97, 2.36, 2.06 folds compared to the Con-group , indicating that TB could activate TLR2/4 at the same time”
  2. Line 327: “1.25, 1.86, 1.58, 1.23 folds, respectively (Figure.3A).” I have corrected these sentences because there were some grammatical errors before. “Interestingly, the MyD88 content in RAW264.7 cells treated with TB was correlated with the expression of TLR2/4. Compared with the control group, 62.5, 125, 250, 500 μg/mL TB increased the expression of MyD88 by 25, 1.86, 1.58, 1.23 folds (Figure.3A).”
  3. lager” is the wrong word, we modify it “large”.
  4. ​ Line 461: “whether TB, like LPS, causes...”? There are some grammatical errors which are not well understood, for which we have made corrections and explanations.

“​One thing to worry about is whether TB may cause excessive activation and inflammation of RAW264.7 such as LPS.” From the point of view of the immune-enhancing effects of TB, the fact that TB can induce cell production of NO and cytokines, while LPS can also induce cell production of large amounts of NO and cytokines, raises the concern as to whether TB can also act as a cellular inflammatory stimulus, as LPS does. Therefore, we have performed many experiments to confirm that TB not only does not cause excessive inflammation in cells, but also further suppresses LPS induced cell inflammation. Thus, TB is a substance that improves immune function and inhibits inflammation.

The above is my response to your comments. I must again thank you for your generous help.

Best Wishes!

Dr. Lei Zhao
